# Carbon Nanofibers Grown in CaO for Self-Sensing in Mortar

**DOI:** 10.3390/ma15144951

**Published:** 2022-07-15

**Authors:** Lívia Ribeiro de Souza, Matheus Pimentel, Gabriele Milone, Juliana Cristina Tristão, Abir Al-Tabbaa

**Affiliations:** 1Department of Engineering, University of Cambridge, Cambridge CB2 1PZ, UK; gm683@cam.ac.uk (G.M.); aa22@cam.ac.uk (A.A.-T.); 2Instituto de Ciências Exatas e Tecnológicas, Campus de Florestal, Universidade Federal de Viçosa, Florestal 35690-000, Brazil; matheus.pimentel@ufv.br (M.P.); juliana@ufv.br (J.C.T.)

**Keywords:** carbon nanofibers, self-sensing, CVD, piezoresistivity

## Abstract

Intelligent cementitious materials integrated with carbon nanofibers (CNFs) have the potential to be used as sensors in structural health monitoring (SHM). The difficulty in dispersing CNFs in cement-based matrices, however, limits the sensitivity to deformation (gauge factor) and strength. Here, we synthesise CNF by chemical vapour deposition on the surface of calcium oxide (CaO) and, for the first time, investigate this amphiphilic carbon nanomaterial for self-sensing in mortar. SEM, TEM, TGA, Raman and VSM were used to characterise the produced CNF@CaO. In addition, the electrical resistivity of the mortar, containing different concentrations of CNF with and without CaO, was measured using the four-point probe method. Furthermore, the piezoresistive response of the composite was quantified by means of compressive loading. The synthesised CNF was 5–10 μm long with an average diameter of ~160 nm, containing magnetic nanoparticles inside. Thermal decomposition of the CNF@CaO compound indicated that 26% of the material was composed of CNF; after CaO removal, 84% of the material was composed of CNF. The electrical resistivity of the material drops sharply at concentrations of 2% by weight of CNF and this drop is even more pronounced for samples with 1.2% by weight of washed CaO. This indicates a better dispersion of the material when the CaO is removed. The sensitivity to deformation of the sample with 1.2% by weight of CNF@CaO was quantified as a gauge factor (GF) of 1552, while all other samples showed a GF below 100. Its FCR amplitude can vary inversely up to 8% by means of cyclic compressive loading. The method proposed in this study provides versatility for the fabrication of carbon nanofibers on a tailored substrate to promote self-sensing in cementitious materials.

## 1. Introduction

Self-sensing concrete refers to concrete materials and structures possessing intrinsic properties that sense various physical and chemical parameters. This property can be harnessed for traffic monitoring [1,2] and structural health monitoring (SHM), including monitoring of load [3], strain sensing [4,5], crack formation [6], freeze-thaw [3], electric magnetic shielding [7] and self-healing performance [8]. To achieve sensing, functional fillers are distributed in the cementitious matrix in order to promote electrically conductive properties [9]. The resultant self-sensing composite is easily prepared and offers good compatibility with concrete structures, making it a more attractive alternative to conventional sensing devices [5]. Among the different types of conductive fillers, steel and carbon materials are commonly used, as recently discussed in several reviews [9,10]. Although the macroscale of the former allows for easier production and use, as well as a lower cost, its main drawback is associated with its susceptibility to corrosion [11]. Carbon nanomaterials, on the other hand, offer high electrical conductivity and durability [9]. Graphene nanoplatelets [12,13], carbon nanotubes (CNTs) [14,15,16] and carbon nanofibers (CNFs) alone, as well as the synergic effect of the combination containing these materials [17,18,19], are examples of conductive fillers used to promote the electrical properties of cementitious systems. Beyond increasing conductivity, the inclusion of carbon-based nanomaterials in cementitious matrices has been gaining attention due to its contribution to accelerating hydration [20] and its effect on durability and mechanical properties [21]. 

Percolation theory is used to describe the electrical behaviour of a system containing a conductive filler. Above a critical concentration, also known as the percolation threshold, these functional fillers form a conductive network inside the matrix [7], leading to changes in the electrical properties as external forces change. Nanofibrous materials, such as CNFs/CNTs, have very high aspect ratios and specific surface areas, resulting in percolation at lower concentrations than for other functional fillers [22]. However, because of the high van der Waals forces between the materials, the high aspect ratios and large surface areas of CNTs/CNFs lead to agglomerations. When dispersed in the cementitious matrix, these functional fillers are likely to cluster rather than be uniformly dispersed [23,24]. As a result, poor dispersion increases the amount of material required to achieve adequate conductivity in the sample [8]. To address this issue, a combination of sonication and surfactants is typically used to increase the dispersion of the nanomaterials. However, sonication is a time-consuming and energy-intensive step that poses a significant challenge in large-scale applications. Experimental work has revealed that cement paste containing 2% CNF by weight of cement is sensitive to its own structural damage [4]. In mortar, xperiments and modelling of the dispersion of CNF in mortar indicate a threshold value between 1.6–2%vol [25,26]. However, previous research also demonstrates a significant decrease in resistivity for cement paste containing 0.1% CNF, as well as changes in resistivity under compressive loads of up to 5% [27]. When comparing CNT and CNF, a smaller fractional change in resistance has been observed when comparing 0.6 wt% of CNF and CNT under identical loading conditions [28]. Thus, the exact values associated with the percolation threshold are unknown, with significant factors associated with the size and length of the CNF, the cementitious matrix used (cement paste or mortar), the water-to-cement ratio and the superplasticizer content used in the mixing [9,29]. To facilitate substrate dispersion in the hydrophilic cement paste, we propose using CaO as a substrate for the growth of CNF, resulting in an amphiphilic composite.

Chemical vapour deposition (CVD) offers a versatile route to obtaining carbon nanofibers in the laboratory. In this process, a vaporised carbon source is catalytically reduced and deposits itself on a substrate as graphite and CNTs/CNFs [30,31]. Catalytic CVD has also been used for large-scale commercial production of multiwalled carbon nanotubes (MWCNT) [32,33]. In this case, the price for MWCNTs is in the range of 0.1–0.15 €/g, depending on the material quality and acquired quantities [34]. In addition to its low cost and scalability [35], an attractive feature of this method is the wide variety of carbon sources, catalyst particles and substrates that can be used for the production of CNTs/CNFs. Thus, the synthesis can be fine-tuned towards sustainable sources of carbon [36], maximising production of CNTs/CNFs by catalyst particle selection and substrate selection based on the final application. For example, in order to facilitate dispersion in the cementitious matrix, carbon nanofibers have recently been synthesised using Portland cement as a substrate [37]. In this case, iron (III) oxide (Fe_2_O_3_) naturally present in the cement acts as a catalyst for the growth of carbon nanotubes. However, the limited amount of catalyst leads to small amounts of carbon nanofibers being produced—typically 2.5 to 3.2% by weight [37,38]. Alternatively, CVD results using Portland as a substrate and containing conversion powder as a catalyst have shown CNT/CNF concentrations as high as 12% by weight [39,40], while using Ni as the catalyst resulted in the production of composites with ~25 wt% of CNT [2]. Due to the high natural iron content, fly ash has also been investigated, using ferrocene as an extra catalyst, yielding CNF ~33% by weight [41]. Furthermore, CNT synthesised on the surface of fly ash showed good dispersion in the mortar and excellent piezoresistive response [42]. Based on a similar principle, we suggest the use of CaO as a substrate for the production of CNF, since the material has an amphiphilic character—hydrophobic CNF grows on top of a hydrophilic substrate [43]. In addition, it offers an excellent bond to the cement and a route to easily remove the substrate by acidic attack, increasing the dispersion of the carbon nanofibers.

Here, we report the production of amphiphilic carbon nanofibers (CNF) and, for the first time, its application to increase the conductivity and promote the piezoresistivity of mortar. We produce CNF using chemical vapour deposition (CVD) of ethanol over a calcium oxide substrate doped with iron nanoparticles. The width and length of the CNF are characterised using SEM, as well as the deposition of the fibres on the surface of CaO. The encapsulation of iron nanoparticles inside the fibres was visualised using TEM. Thermogravimetric analysis of the produced fibres is used to investigate the thermal stability and concentration of the CNF over the CaO substrate. Once the CNF@CaO is dispersed in the matrix, SEM is also used to investigate the dispersion and interaction of the composite. To demonstrate the role of the CNF in decreasing the electrical resistivity of mortar, different concentrations of fibres are dispersed in the matrix and the conductivity is measured using the four-point probe method. To assess the piezoresistive response of the material, a cyclic compressive load is applied and the gauge factor quantified. The approach in this study demonstrates versatility for fabrication of carbon nanofibers with a tailored substrate to achieve better dispersion of the nanofibers in the cementitious matrix. 

## 2. Materials and Methods

Magnetic carbon nanofibers supported in CaO (CNF@CaO) were produced via chemical vapour deposition using ethanol as a reagent. First, mixed calcium iron oxide was produced via wet impregnation by adding 0.8 g of calcium oxide to an aqueous solution containing 1.4 g of iron (III) nitrate to produce samples with Fe/CaO contents of 20 wt%. The resulting suspension was heated to 90 °C, under constant stirring, until complete evaporation of the water. Then, the material was calcined at 800 °C for 1 h. The resulting substrate is composed of Ca_2_Fe_2_O_5_, as indicated in Equation (1). The presence of this phase was confirmed in previous work [43], where XRD and Mӧssbauer spectroscopy confirmed the presence of Ca_2_Fe_2_O_5_. The presence of mixed iron oxide, instead of phases of stable oxides such as hematite, indicates that the iron was homogenously added to the substrate [43].
(1)$$2\mathrm{Fe}{\left({\mathrm{NO}}_{3}\right)}_{3}\cdot 9H_{2}O+2\mathrm{CaO}\;+{\;O}_{2}\to {\mathrm{Ca}}_{2}{\mathrm{Fe}}_{2}O_{5}+6{\mathrm{NO}}_{x}+18H_{2}O$$

After calcination, a nitrogen flow of 100 mL min^−1^ was purged through a gas-washing bottle containing ethanol at room temperature. The resulting nitrogen, saturated with ~6 vol% of ethanol, passed through a quartz tube containing 200 mg of mixed calcium iron oxide. The quartz tube was placed horizontally inside a furnace (Tubular furnace 1200, Sanchis, Porto Alegre, Brazil) and the temperature was raised at a heating rate of 5 °C min^−1^ up to 900 °C and maintained for 1 h. The production of carbon nanofibers is shown schematically in Figure 1. The iron nanoparticles dispersed in CaO acted as catalysts for the dissociation of the gaseous phases and templates for the nucleation and growth of the carbon nanofibers [44]. After CVD, Mӧssbauer analysis was performed in the as-synthetised samples to identify the iron phases after the reduction with ethanol. Phases of α-Fe and iron carbide were primarily identified (75%) [43]. Other phases were associated with CaFe_2_O_4_ and a solid solution of iron and carbon (γ-Fe(C)). XRD showed different phases of CaCO_3_, Ca(OH)_2_, CaO and Fe formed. The calcium oxide substrate could be easily removed after acid attack, but also acted as a hydrophilic moiety, conferring amphiphilic properties to the CNF@CaO composite. In this work, to remove the substrate, 2.6 g of CNF@CaO as grown was mixed with 50 mL of HCl 1 M. After mixing, the CNF was separated with a magnet until the solution was clear. Then the acid was removed and the CNF was washed with water until neutralisation. 

Scanning electron microscopy (SEM) was carried out using a Evo LS15 (Carl Zeiss, Cambridge, UK). The samples were gold coated using a rotary pumped sputter coater (Agar Sputter Coater B7367A, Agar Scientific Ltd., Essex, UK) to improve the conductivity of the surface of the samples and to prevent overcharging. For the CNF deposited in the surface of CaO, the SEM operated at an accelerating voltage of 8 kV; for the mortar composite containing CNF, the voltage was 6 kV. It was difficult to distinguish ettringite from nanofibers in the overall surface sample at the maximum distance in the SEM. As a result, the air bubbles in the mortar were chosen for investigation because they contained small agglomerates of fibres. To measure the diameter and length of the CNF, as well as observe the encapsulation of iron nanoparticles and graphene layering of the carbon nanofibers, a transmission electron microscope was used (TEM, FEI Tecnai Osiris FEGTEM, Hillsboro, OR, USA). The CNF@CaO samples were washed with acid, followed by water neutralisation. The suspended material was then placed over a carbon film with a Cu grid for analysis.

Thermogravimetrical analysis (TGA) was used to assess the weight ratio of as-grown CNF@CaO using a PerkinElmer STA6000 (Shelton, CO, USA) between 100 and 800 °C at a rate of 5 °C min^−1^ under air atmosphere. After acid washing the material with HCl 1 M, followed by washing with water until the pH was ~5, another TGA was performed of the dried material in the same conditions as before. Approximately ~5 mg of material was analysed. To investigate the magnetic properties of the samples, the hysteresis of CNF@CaO and washed CNF were measured using a Lakeshore Cryotronics (Westerville, OH, USA) vibrating sample magnetometer as a function of the magnetic field up to 15 kOe at room temperature. Raman spectra were collected on a XploRA Plus (Horiba Jobin Yvon, Villeneuve d’Ascq, France) using a 638 nm laser. 

Ordinary Portland Cement (CEM I 42.5) provided by Heidelberg-UK was used for the production of the mortar samples. Sieved and cleaned sand ranging between 0.18–2 mm was used as fine aggregate. The workability of the mixture was aided by adding 0.3 wt% by weight of cement (bwoc) with modified polycarboxylic ether as the superplasticiser (MasterGlenium 315C, BASF). All mortar mixes had a water–cement ratio (w/c) of 0.6 and a sand–cement ratio (s/c) of 3. In situ grown carbon nanofibers were added to the mortar matrix at 0.4, 1.2 and 2 wt% by weight of cement (bwoc) or 0.36, 1.08 and 1.8% by volume of cement, respectively, considering a density of CNF ~1.6 g/cm^3^ and the cement density as 1.44 g/cm^3^. The quantity of CNF@CaO to achieve the mentioned concentrations of carbon nanofibers was calculated with the weight ratio obtained from TGA. To assist in the dispersion of carbon nanofibers, the CNF@CaO composite was sonicated (Fisherbrand FB11203, Singer, Germany, 80 kHz frequency and 100% power) for 30 min in a solution of water and superplasticiser. The ready-dispersed CNT and superplasticizer suspension were mixed into a rotary mixing according to the following protocol: sand and cement were dry-mixed in a mixer pan for 3 min, and then half of the water containing the superplasticiser and the nanomaterials was added and mixed in for another 1 min, before adding the rest of the water and mixing for another 1 min. The mixture was then mixed at maximum velocity for 0.5 min, followed by 1 min at minimum speed before adding to the oiled mould. The process is schematically represented in Figure 2.

The mix proportions are shown in Table 1. Triplicates of all samples were cast into oiled moulds to produce samples of 20 × 20 × 80 mm. Four perforated steel sheets of dimension 40 × 17 × 0.55 mm, with a 3 mm hole diameter, were embedded in the sample (Figure 3A). All samples were placed in an electric vibrator (Controls Automatic Sieve Shaker D407, Cernusco, Italy) for 30 s for good compaction and to reduce air bubbles, then covered with plastic film for 24 h, curing at room temperature. The specimens were demoulded and cured in a moist container at 20 ± 1 °C and with a relative humidity ≥95% for 28 days.

The electrical resistance (R) was measured using the four-probe method with a digital multimeter (TTi 1604, Aim & Thurlby Thandar Instruments, Huntingdon, UK), as shown in Figure 3B. Direct current (DC) of 20 V was applied between the two outer electrodes and the electric potential was measured between the two inner electrodes (Figure 3B). An insulator was also added in between the compressive plates and the sample. The electrical resistivity (*ρ*) was calculated using the following equation:(2)$$\rho =R\cdot \frac{A}{L}$$
where *ρ* is electrical resistivity in ohm meters, *L* is the internal electrode distance in meters, *A* is the electrode area in square meters and *R* = V/I is the measured resistance determined by measuring the voltage drop across the specimen (V) in volts and the applied current (I) in amperes. To tackle the polarization effect, the values were collected after 15 min of constant tension applied to the outer two electrodes [45]. The application of compressive load to the mortar sample was accomplished with a hydraulic press (Instron 5567, Norwood, MA, USA—30 kN capacity) operating under a distance control of 0.8 mm min^−1^, up to 7.5 MPa. Figure 3C shows a schematic of the PC-controlled acquisition system that was used to collect the data for load, strain gauge, voltage and current from the samples while a voltage of 20 V was applied.

## 3. Results

### 3.1. Production and Characterisation of Carbon Nanofibers

The carbon nanofibers (CNF) containing magnetic nanoparticles were synthesised using chemical vapour deposition (CVD) with CaO as the substrate. Calcium oxide was selected as a substrate due to its compatibility with cementitious materials and its polar nature, which grants an amphiphilic property to the CNF@CaO composite. Through CVD, the substrate containing mixed calcium iron oxides as catalyst reacted with nitrogen saturated with ethanol at 900 °C. At this temperature, the reduction of Ca_2_Fe_2_O_5_ by ethanol results in the formation of Fe^0^, Fe_3_C, CaFe_2_O_4_ amorphous carbon and CNF [43]. The production of carbon nanofibers is shown schematically in Figure 1 and Figure 4A,B shows entangled CNF grown on the surface of CaO. 

The calcium oxide substrate was then removed by acid washing, resulting in free CNF to be observed using TEM and shown in Figure 4C–I. Figure 4C shows the different widths of the CNF, with fine CNF around 40 μm and typical ones ~160 μm. The high length-to-width ratio of the CNF is demonstrated in Figure 4D, with fibrous structures ~11 μm long and 280 μm wide. In general, the length of the CNF ranged between 5–10 μm and the diameter was between 40–300 nm, with average values around 160 nm. The high length-to-diameter aspect ratio of the CNF is particularly suitable for sensing, as it contributes to more conduction paths [9]. Amorphous carbon was observed through TEM analysis, as shown in the central part of Figure 4D. After the CNF has grown, the catalyst particles are trapped inside the graphene layers, as shown in Figure 4E, and empty channels are inside the CNF (Figure 4F). The angle between the graphite basal planes and the tube axis is different from zero, resulting in the term carbon nanofibers of composites. In contrast, the term carbon nanotube (CNT) refers to graphene sheets rolled up in concentric cylinders with walls parallel to the axis [46]. This is demonstrated in Figure 4F–I, where the angle between the nanosheets and the tube axis is different from zero; Figure 4I is a zoom of the graphene layers in Figure 4H. Although the recorded images identify the presence of carbon nanofibers, multi-walled carbon nanotubes are likely to also occur. Controlling the size distribution of the catalyst particles is a standard way to tune the CNF diameter dispersion [44]. As a result, the nanoparticles trapped inside the nanofibers have an elongated shape and vary between 46–200 nm, slightly smaller than the diameter of the CNF. These trapped nanoparticles inside the nanofibers are responsible for the magnetic behaviour of the CNF, even after washing with acid.

Thermogravimetric analysis (TGA) was performed to quantify the metal content after full oxidation of carbon and to examine the thermal stability of the as-grown CNF over substrate. The oxidation temperature was determined by TGA curves (deflexion point—Figure 5A) and more precisely by the DTA curves (Figure 5B). The curves of the as-grown CNF show: (1) oxidation between 372–427 °C, attributed to amorphous carbon, where ~2% of the mass was lost; (2) a second weight loss, attributed to the oxidation of CNF, peaking at 602 °C, where 26% of the mass was lost; and (3) at around 688 °C, loss of 26% of the mass, corresponding to the transformation of CaCO_3_ [43] into CaO. The residual material was ~45%, representing the CaO support and the iron-based catalyst. After acid washing the sample, the CaO was removed, together with most of the amorphous carbon during the neutralisation process. Consequently, the TGA (black curve) presented only one thermal event, corresponding to the oxidation of the CNF, with a maximum weight loss rate at 590 °C, where 84% of the material was lost. The residual mixture of catalyst and any residual support made up ~11%, comprised of catalyst metallic oxides. This is consistent with the previous characterisation of the washed CNF, in which the elemental analysis revealed a carbon concentration of ~80 wt% [43]. At ~580 °C, an anomalous mass gain is observed, where a sharp decrease in specimen mass is accompanied by a rapid increase in temperature by 6 °C, followed by a quick decrease. Similar behaviour has been observed on as-produced, unpurified and uncompacted nanotubes [47] and attributed to spontaneous combustion; i.e., the heat released in the exothermic reaction is enough to sustain rapid burning of the sample [48]. This behaviour seems to be mainly associated with single-walled carbon nanotubes, which could imply their presence in the sample, despite not being observed in the TEM.

The Raman spectra obtained for the pure CNF are shown in Figure 6C (bottom). In the first-order region, two bands can be observed in the red laser (638 cm^−1^). The so-called D band is sited around 1325 cm^−1^, and it is associated with disordered structures in carbon materials [49]. The peak around 1575 cm^−1^ is the G band associated with the high degree of symmetry and order of carbon materials [50]. Finally, a third, weak band is observed at 1614 cm^−1^, associated with D’. The strong, dispersive band around 2643 cm^−1^ is designated as the G’ band (called 2D sometimes). For the sample CNF@CaO (Figure 6C—top), the bands are in a similar region, but much more intense. When comparing the I_D_/I_G_ ratios calculated from the intensities in the D and G bands, a small rise is observed: the ratio for CNF@CaO is 0.57, whereas the ratio for washed CNF is 0.64. These findings suggest that washing the sample increases the ratio of disordered to organised carbon. This could be attributed to defects created in the graphitic structure during the acid washing for the removal of CaO.

To determine the magnetic properties of the CNF samples, a vibrating sample magnetometer was employed at room temperature. As indicated in Figure 6D, CNF and CNF@CaO have saturation magnetisation values of 7.3 and 11.8 emu/g, respectively. Similar values were reported for iron-doped diatomite, which exhibited a saturation magnetisation in the range of 10–15 emu/g [51]; other reports of CNT doped with a high iron content demonstrated a saturation magnetisation as high as 38 emu/g [52]. Coercivity values (H_c_) for pure CNF and CNF@CaO are −414 and −277 Oe, respectively. This low coercivity of remanence is indicative of a superparamagnetic property (i.e., responsiveness to an applied field without retaining any magnetism after removal of the same). This property allows for simple separation of the washed carbon nanofibers suspended in solution. 

### 3.2. Dispersion of Carbon Nanofibers in Mortar

Scanning electron microscopy was used to investigate the dispersion of the carbon nanofibers in mortar, using the sample with 2 wt% of CNF. The highest content of carbon nanofibers was used due to the challenge in locating the carbon nanofibers in the matrix—the carbon nanofibers are difficult to differentiate from hydration products, particularly ettringite. After the CNF was dispersed and the material cured in high humidity for 28 days, the sample was dried in air, followed by vacuum drying before SEM. The fibres were mainly observed as small conglomerates, ranging between 3 and 7 μm, inside the pores in the mortar (Figure 6A–C). No conglomerate larger than 7 μm was observed, indicating the nanofibers are distributed in small nest-like bundles throughout the matrix. Previous investigations of carbon nanofibers and nanotubes distributed in cementitious matrix have shown that a poor dispersion may lead to agglomerated CNT; alternatively, a better dispersion of the fibres leads to high-density products dispersed as clumps [53,54]. Figure 6F also shows portlandite entangled with the carbon nanofibers (Figure 6D,E), demonstrating the hydration of the CaO substrate. 

### 3.3. Influence of CNF Concentration on Electrical Resistivity

The addition of carbon nanofibers to the mortar has the effect of increasing the electrical conductivity of the specimens. The electrical resistivity measured with the four-probe point method is presented in Figure 7 and differed with the hydration levels of the specimen, as well as the content of carbon nanofiber and presence of the CaO support. The electrical resistivity was much lower for samples immediately after 28 days of curing in a high-humidity (>95%) environment, as shown in Figure 7A. For the damp samples, the voltage between the internal electrodes at the central electrodes was ~8–10 V and the current ranged from 400 to 1400 µA. As a result, the electrical resistivity of the mortar samples with high relative humidity was between 80–230 Ω m, with a slight decrease in conductivity for samples with washed CNF and CaO. Similar values were found in water-saturated mortar samples containing CNF [3,8]. These lower values of electrical resistivity to the water-saturated specimen were attributed to the high conductivity of the pore solution. The electrical resistivity of mortar samples after drying (Figure 7B) was markedly increased—between 0.77–42 kΩ m. For the dried samples, the voltage measured in the internal electrodes was ~4–8 V whereas the current was significantly lower—between 1 and 110 µA. Furthermore, for the dried samples, the contribution of the content of CNF to the changes in conductivity was more pronounced when compared with the damp samples. In this case, the samples with control and with 0.4 wt% of CNF in mortar presented similar values of resistivity, ~2 × 10^4^ Ω m. This indicates that, up to 0.4 wt% CNF, the mortar is still presenting DC electrical resistivity in the range of 10^4^–10^7^ Ω m, i.e., acting as an quasi-insulator [9]. However, when the content of CNF increases to 2 wt%, the resistivity drops to 5 × 10^3^ Ω m, which is at least one order of magnitude less than the control. For this sample, an increased standard deviation was observed, possibly associated with the sample’s non-uniform conductivity and bonding issues with the steel electrode. In addition, the values of electrical resistivity were 7.7 × 10^2^ Ω m, two orders of magnitude lower, for the samples containing 1.2 wt% of CNF after the material was washed with acid, i.e., when the CaO was removed. Unfortunately, two specimens containing CNF(A) −1.2 wt% fractured between the matrix and the steel electrode; thus, there was no standard deviation associated with this sample. The decrease in resistivity highlights the strong effect of the CNF in the composite. These results are in agreement with other reports, where the required dosage of CNF to achieve a well-established current through tunnelling varies between 0.6 wt% [28] and 2.25 vol% [23]. The results also show the increased conductivity of the composites once the support is removed. 

Interestingly, the control with only the substrate, i.e., the precursor used for the deposition of the material, was also investigated and the values of resistivity were ~1.1 kΩ m. The substrate was produced from the reaction between CaO and Fe^3+^ to produce mixed oxides with iron and calcium, Ca_2_Fe_2_O_5_, as shown in Equation (1). It is the reduction of Ca_2_Fe_2_O_5_ by ethanol that results in the formation of the carbon nanofibers, and also the reduced forms of iron, including nanoparticles of iron and iron carbide. In this case, ~1.8% by weight of cement (bwoc) of calcium oxide containing mixed oxides with iron and calcium was added to the mortar. Other authors have also investigated the use of nanoparticles of iron oxide to successfully increase the conductivity of cementitious matrix [55]. However, it is interesting to notice that the conductivity of the samples with iron oxides is markedly increased when compared with samples containing more iron and CNF. The reason behind the difference is not immediately apparent. Iron oxides are poorer conductors when compared with the reduced forms of iron [56]. However, it could potentially highlight how the pure powder dispersion in water is a lot more favourable than the material coated with CNF. Therefore, the dispersion is favoured and increases the conductivity. However, more studies are necessary. 

### 3.4. Influence of CNF Concentration on the Piezoresistive Response under Compressive Loading

The piezoresistivity of the mortar samples under compressive load was investigated in the elastic region. This focuses on the potential application of the composite for load detection and structural soundness, i.e., for non-destructive systems. The four electrodes in the mortar samples were positioned perpendicular to the plane of compression (Figure 3a), with a load exerted at a rate of 0.8 mm min^−1^ up to 7.5 MPa. The linear response of the stress–strain curve aspect (Figure 8) confirms the elastic behaviour of the sample and the slope of the curve was 29 GPa for the control sample with SP (CTRL-SP), 42 GPa for the control sample with CaO (CTRL-CaO) and between 23.7 and 35.3 GPa for the samples with CNF. The addition of CNF/CNTs in cementitious composites has been proven effective in developing the mechanical properties of the materials [57], since fibres can bridge microcracks, fill pores and accelerate hydration. However, the obtained stiffness parameters are not accurate due to the presence of electrodes and the cuboid shape of the samples. Compression tests following standards were not performed in this study due to the limited amount of material.

Mortar samples containing CNF showed piezoresistive behaviour dependent on the concentration of CNF. To investigate the stability and repeatability of the piezoresistive response to compressive loading, 18 cycles with an amplitude of 7.5 MPa and a loading rate of 0.8 mm·min^−1^ were applied to the mortars. The fractional change in resistivity (FCR) was calculated by dividing the difference in resistivity at each point with the no-load resistivity (or baseline resistivity). Figure 9 shows that the electrical resistivity of all mortars is consistent with expected response under compression and decreases with the increase in the compressive load due to the shortening of the conduction path. Typically, samples take some time to stabilise, with the first few cycles still not returning to zero. The increase in the baseline electrical resistivity over time is caused by the polarisation effect, mainly due to the presence of water and dissolved ions in the water. This also indicates that residual water is still present in the sample, as dry samples are less prone to variations on the curve [27]. Furthermore, micro damages separating adjacent nanofibers may also lead to an increase in resistivity, consequently, an increase in the baseline [6]. The control mortar without CNF (Figure 9A), as well as the sample with only CaO and mixed iron and calcium oxides (Figure 9B), present a small amplitude for the FCR values on loading, with a maximum modular FCR (FCR_max_) of 1%, indicating a negligible piezoresistive behaviour. The FCR of the samples with CNF reached FCR_max_ of 0.5, 8.4 and 1.3% in concentrations of 0.4, 1.2% and 2 wt% CNF, respectively. For the sample with washed CNF at 1.2 wt%, the FCR_max_ was 1.0%. 

The obtained values, as well as the pronounced change in resistivity for the sample with 1.2 wt% of CNF, were interpreted in relation to percolation theory [9]. For the samples with CNF 0.4 wt%, we conjecture that the concentration may not be sufficient to achieve a conductive network capable of sensing the variation in the state of strain. Thus, values of resistivity (Figure 7) and FCR under compressive loading were similar to the control. On the other hand, when the CNF@CaO concentration increases to 1.2 wt%, changes in contact resistance increase due to the formation/breakage of CNF junctions. In addition, under compressive loading, the distance between CNF decreases, facilitating a tunnelling effect. The unique nest-like morphology of the CNF (Figure 6D,E) distributed in the mortar may also facilitate the high FCR observed in this sample, providing many possible locations for triggering CNF contact and tunnelling [42]. When the concentration of CNF increases further, by the addition of 2 wt% of CNF and 1.2 wt% of washed CNF, the sample already has a more pronounced and stable conductive network. Thus, its electrical properties were not particularly affected by any external load; that is, the piezoresistive effect caused by the variation in proximity between CNFs is not so pronounced. As a consequence, the resistivity values (Figure 7) were lower than the control, but the FCR_max_ is similar to the control. This indicates that, to obtain a piezoresistive behaviour for the sample with washed CNF, reduced concentrations of material should be used. When compared to the literature, a maximum FCR of 9% was observed for carbon nanofibers at 2.25 vol% [23] and an FCR maximum of 2% for mortar samples with 2.5 vol% of CNF [58], thus indicating a lower concentration of the material here also shows a good result for the sample at 1.2 wt% (1.08 vol%). Alternatively, an FCR max of 5.5% was observed for the cement samples with 0.1 wt% of CNF, indicating a better piezoresistive response under loading [27]. This could be attributed to a conductive network being obtained in this sample under compressive loading at 0.1 wt% of CNF. It could be that the dispersion of the material is better, since the material does not form clumps. This could have been the case for washed samples if the CaO substrate had not been used. 

To examine the relationship between strain and variations in electrical resistivity in the mortar, the strain versus FCR curves are presented in Figure 10. To quantify the strain sensitivity, the gauge factor (*GF*) was used, as it represents the relative change in electrical resistivity due to the mechanical resistance. The relationship between strain amplitude and resistance change can be described as follows:(3)$$GF=\frac{d\rho /\rho }{\varepsilon_{1}}$$
where *ε*_1_ is compressive strain measured by a strain gauge and *d**ρ*/*ρ* is equal to the FCR. The *GF* is then obtained by applying Equation (3) via fitting with a linear regression to the FCR–strain curve, as shown in Figure 10. 

For the mortar without CNF, as well as the mortar with 0.4, 2 wt% of CNF and 1.2 wt% of washed CNF, the GF is ~30, confirming that the specimens are not applicable for strain sensing under a compressive load. However, when 1.2 wt% of CNF is added, the GF markedly increases to 1552, demonstrating the high sensitivity of the material. Moreover, the sample with 1.2 wt% shows a more constant variation, without noise. This indicates the potential of mortar embedded with CNF for more stable piezoresistive behaviours. Recently, Ding et al. (2022) investigated the piezoresistivity of cement composites containing 0–5 wt% of CNT synthesised in the surface of cement, resulting in gauge factors of 22–748 [2]. Likewise, CNT@cement embedded in mortar at a concentration between 0.4–2 wt% resulted in gauge factors between 744 to 1170 [42]. For future work, a different design of experiments could be considered, including more points around 1.2 wt% CNF, to better understand the behaviour of the material. 

## 4. Conclusions

This study presents a versatile method for fabricating carbon nanofibers on substrates designed to facilitate the dispersion of the nanomaterial. Using iron as a catalyst, CNF was produced by chemical vapour deposition (CVD) on the surface of calcium oxide (CaO). This substrate was chosen due to its ease of removal, which results in free CNF, and its polarity, which lends an amphiphilic quality to the composite and facilitates its dispersion in mortar. Small agglomerates of loose fibres were observed with a SEM after the fibres were dispersed in the mortar, indicating a good distribution of the material. Increasing the concentration of CNF in the mortar resulted in a decrease in resistivity, with the lowest values occurring at around 2 wt%. Additionally, after removing CaO, the electrical resistivity decreased to 0.8 kΩ m for contents containing approximately 1.2 wt% CNF. Under compressive loading, the piezoresistive response of CNF was studied, and the composite containing 1.2 wt% of unwashed CNF exhibited an excellent variation in electrical resistivity. The gauge factor (GF) was used to quantify the sensitivity to deformation, and the sample containing 1.2% by weight of CNF@CaO had a gauge factor of 1552 while the others had gauge factors below 100. This sample’s exceptional deformation sensitivity suggests that the contact points formed between small adjacent CNF@CaO clusters can be easily formed and broken, thereby increasing the nanocomposite’s sensitivity. Potential applications for the enhanced electrical properties include evaluating the condition of civil engineering structures.

## Figures and Tables

**Figure 1 materials-15-04951-f001:**
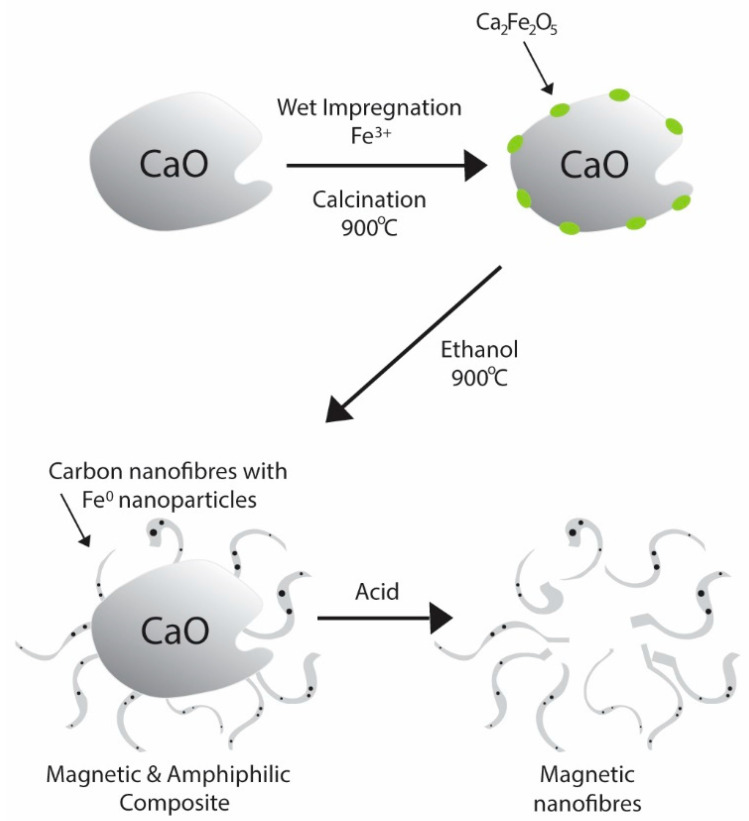
Schematics of production of mixed calcium iron oxide, carbon nanofibers supported in CaO and pure magnetic nanofibers.

**Figure 2 materials-15-04951-f002:**
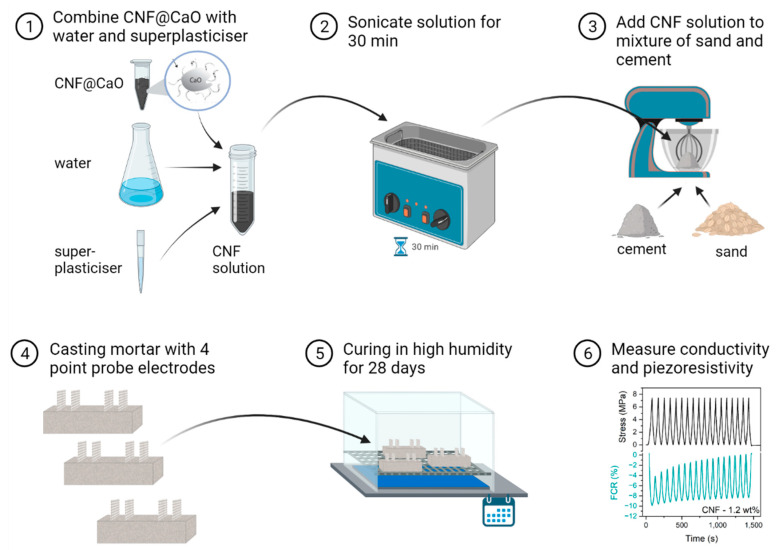
Workflow used for dispersion of CNF in mortar, followed by measurement of electrical properties.

**Figure 3 materials-15-04951-f003:**
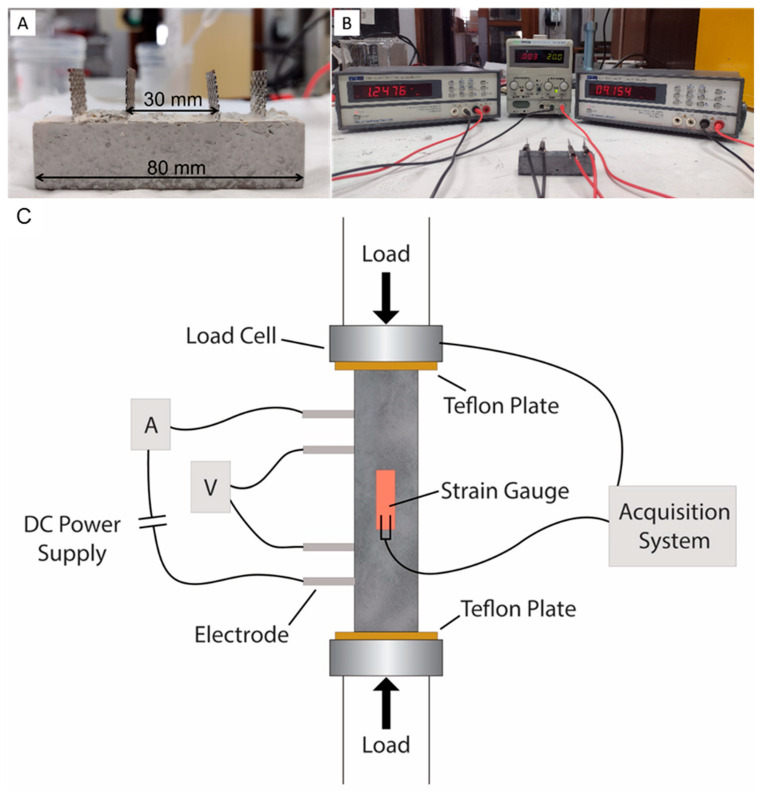
Set-up for measuring the conductivity of the samples. (**A**) Mortar sample with 4 electrodes; (**B**) multimeters for measurements; (**C**) experimental set-up used to measure the piezoresistive behaviour of the mortar samples containing CNF.

**Figure 4 materials-15-04951-f004:**
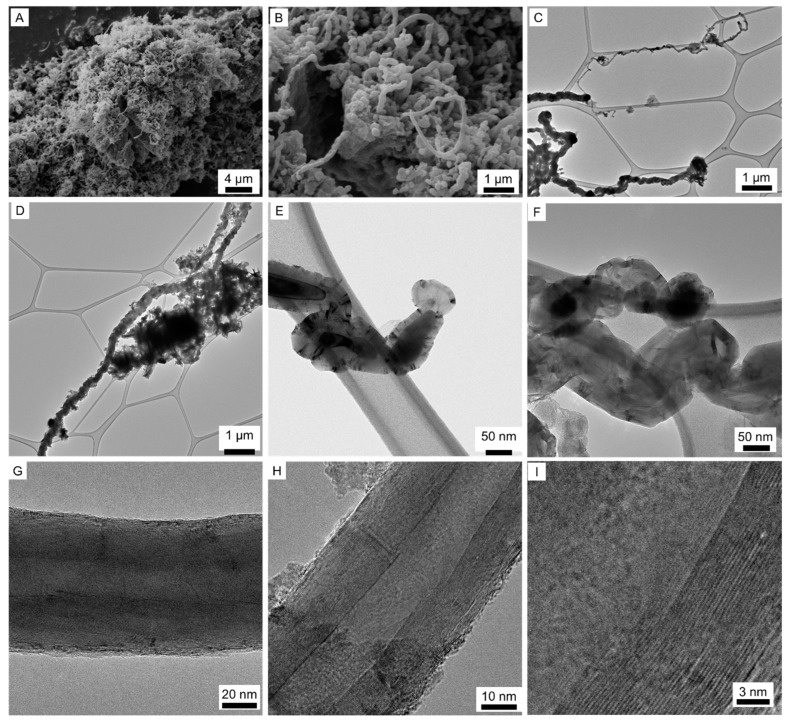
Carbon nanofibers supported in calcium oxide. (**A**,**B**) SEM images from the as-grown samples showing the growth of the CNF on the surface of calcium oxide particles. (**C**–**I**) TEM images after the acid washing for the removal of the calcium oxide support.

**Figure 5 materials-15-04951-f005:**
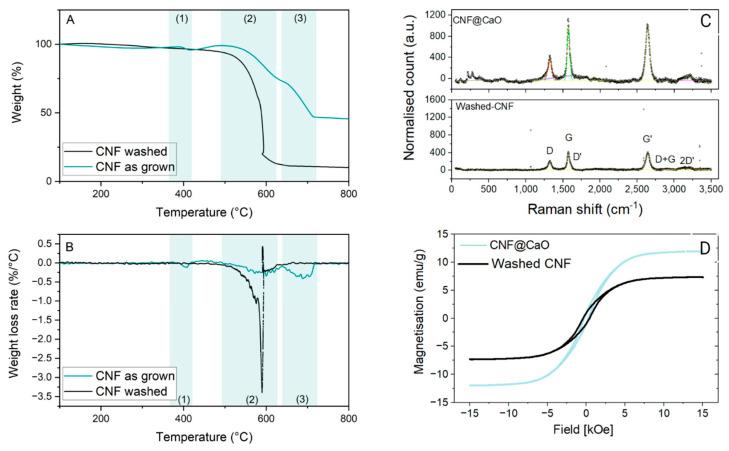
Characterisation of CNF@CaO as produced and CNF after acid washing. (**A**) Thermogravimetric analysis of as-grown CNF in the surface of CaO (teal) and acid-washed nanomaterial, purified for the removal of CaO (black); (**B**) Weight loss rate for the as-grown CNF (purple) and purified CNF (teal); (**C**) Raman spectra of CNF@CaO (top) and CNF—washed (bottom); (**D**) Magnetisation curve of CNF@CaO (teal) and CNF—washed (black).

**Figure 6 materials-15-04951-f006:**
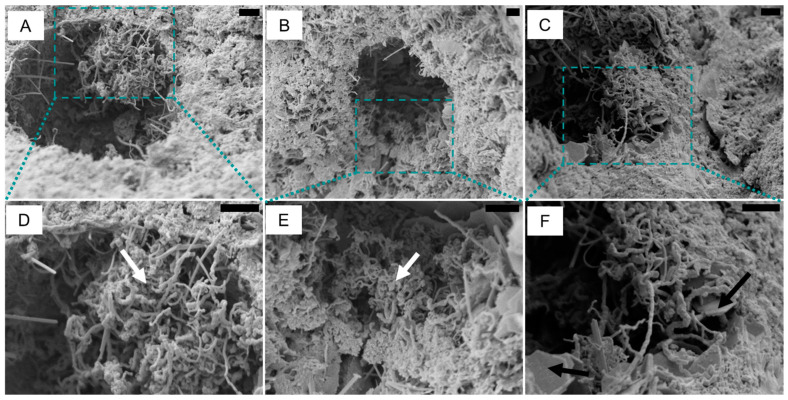
Scanning electron microscopy of carbon nanofibers supported in CaO, dispersed in mortar. Scale bar 2 μm. (**A**–**C**) mortar surface containing the nest of CNF; (**D**–**F**) details of the CNF next, white arrows indicate CNF and black arrow indicate portlandite.

**Figure 7 materials-15-04951-f007:**
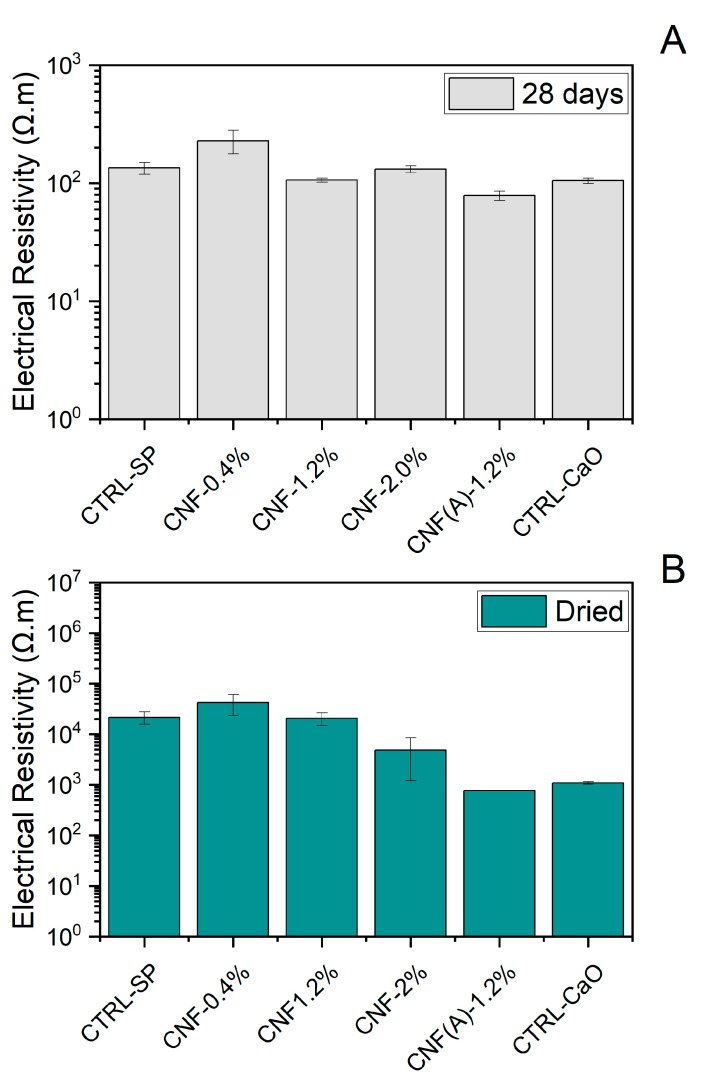
Effect of CNF concentration on the electrical resistivity of mortar specimens (**A**) immediately after 28 days in a high-humidity chamber and (**B**) after drying.

**Figure 8 materials-15-04951-f008:**
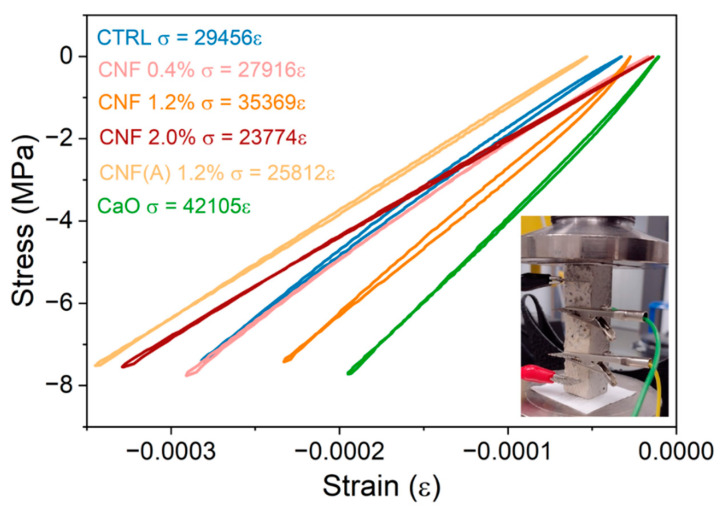
Stress–strain relationship of mortar specimen under compressive loading. Inside detail: experimental setup for the piezoresistivity testing.

**Figure 9 materials-15-04951-f009:**
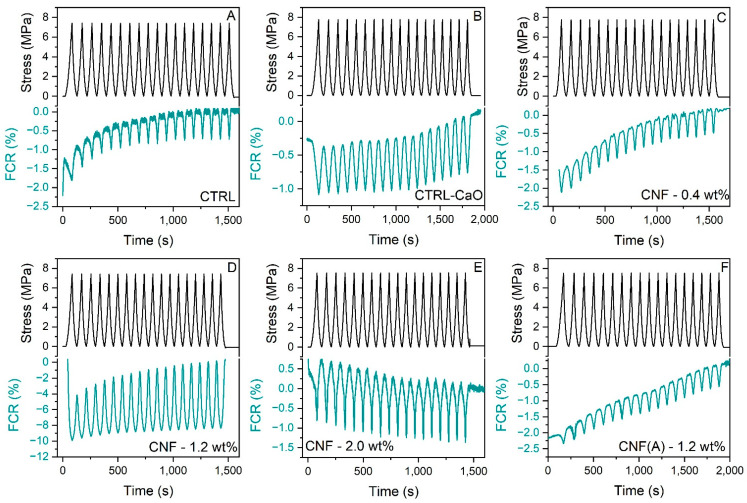
Piezoresistive behaviours of mortars under cyclic compressive loading. (**A**) Control mortar; (**B**) control mortar with CaO substrate; (**C**) mortar with 0.4 wt% of CNF; (**D**) mortar with 1.2 wt% CNF; (**E**) mortar with 2.0 wt%; (**F**) mortar with 1.2 wt% washed CNF.

**Figure 10 materials-15-04951-f010:**
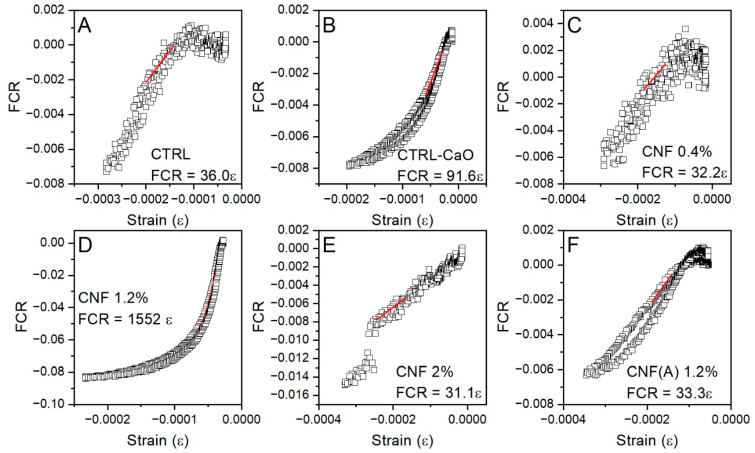
FCR versus strain for mortar specimens at different CNF concentrations under a monotonic uniaxial compressive loading with an amplitude of 7.5 MPa. (**A**) Control mortar; (**B**) control mortar with CaO substrate; (**C**) mortar with 0.4 wt% of CNF; (**D**) mortar with 1.2 wt% CNF; (**E**) mortar with 2.0 wt%; (**F**) mortar with 1.2 wt% washed CNF.

**Table 1 materials-15-04951-t001:** Mix proportions used for test mortars.

Specimen	w/c	s/c	CNF (% bwoc)	CNF@ CaO (% bwoc)	Superplasticiser (% bwoc)
Control	0.6	3	-	-	0.3
CNF 0.4	0.6	3	0.4	1.6	0.3
CNF 1.2	0.6	3	1.2	4.8	0.3
CNF 2	0.6	3	2.0	8.0	0.3
CNF 1.2(A)	0.6	3	1.2	-	0.3
CTRL-CaO	0.6	3	-	1.9	0.3

## Data Availability

The raw/processed data required to reproduce these findings cannot be shared at this time as the data also forms part of an ongoing study.

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
