# Peer review of "Carbon Nanofibers Grown in CaO for Self-Sensing in Mortar"

_materials, 2022, doi:10.3390/ma15144951_

Round 1
Reviewer 1 Report
The paper describe the realization of mortar based nanocomposites filled with CNF or CNT produced through CVD methods. The idea to grow carbon nanotubes/nanofibers on the surface of CaO particles in order to obtain amphiphilic particles, easier to disperse in the matrix, is interesting and well developed. The characterization is well exposed and designed. I think that the paper can be published after minor revisions, described below:
· Table 1: what is the difference between CNF 1.2 and CNF 1.2 (A); if they are the same sample I suggest to remove on of them by the table because can cause misunderstandings;
· Section 3.1: correct all the figure citations; for example, text refers to figure 3 (A-B) that doesn’t exist;
· Figure 4 caption: Pictures E and F are not described in the caption;
· Page 7, row 22: Where authors mentions CaCO3? the substrate for the CNT/CNF growth should be CaO; could authors explain why there is CaCO3 as substrate;
· Section 3.4: correct all the figure citations.
Reviewer 2 Report
The authors presented a method for synthesis of CNFs which uses CaO as a substrate for CNFs growth by CVD technique. Then they embedded the resultant CNFs into mortar structure that improved the self-sensing properties of matrix. The idea is interesting and I recommend the paper for publication after some corrections.
The following comments could be helpful to improve the manuscript:
1. It’s better to mention the values of key parameters in CNFs synthesis method. The prominent factors in CVD growth are temperature, pressure, feed composition and growth time.
2. As discussed in the “introduction section”, other carbon nanomaterials could be used as the filler in such a composites. For example, graphene has higher electrical conductivity than other carbon materials. Also, graphene has other superior properties even better than CNFs. Why did the authors use CNFs? The advantages of this material over other ones should be add to “Introduction”.
3. In the “Introduction”, the authors said that the sonication method is not a large-scale method for dispersing the filler in the matrix material. But, in the next paragraph, CVD method introduced as an experimental technique for synthesis of CNFs. Could the CVD method be used for large scale production? It seems the answer is no. If so, could the authors just propose an alternative route for CNF synthesis to make the idea promising for large-scale production?
4. The explanation about the analysis of materials is not necessary to be in the last paragraph of “Introduction”. The last paragraph should contain the novelty of the work.
5. How did the authors choose the range of filler content (0.4-2%)? Any pre-test was performed? According to the enhanced electrical resistivity of composite with the low filler content of 0.4%, why the lower percentage did not used to reach the possible better performance? And for the optimal piezoresistive behaviors (1.2% CNFs), it would be better to have more filler compositions to have the optimum content. Design of experiments could be helpful in these cases. But, now some more points around the reported optimum (1.2%) could get the optimal value.
Reviewer 3 Report
The work in the article is good. But authors have not read the article before submitting it. Improper figure number citations in entire manuscript. Authors need to address following points to improve the quality of the article.
· Add Keywords
· “Where ρ is electrical resistivity, L is the internal electrode distance, A is the electrode area, R=V/I is the measured resistance determined by measuring the voltage drop across the specimen (V) and the applied current (I).” Add the SI units for all the indicators.
· Figure 2 is not properly cited in the text.
· “The electrical resistance (R) was measured using the four-probe method with a digital multimeter (TTi 1604), as shown in Figure 2B.” Is it figure 2B or 3B?
· Figure 4 (A-F) is wrongly numbered as Figure 3(A-F) in Text. Correct the same.
· “Figure 5F also shows port-landite entangled with the carbon nanofibers,” There is not Figure 5F. Figure 5A & B is not cited properly in the text.
· A strong advise to authors that they need to cross check the citation of all figures in the text for the entire manuscript with correct numbering.
· Figure 6 is not cited and missing the proper explanation.
· In figure 7B, why is error graph of CNF-2% high? Why there is not error graph for CNF(A)-1.2%?
· Results looks very shallow as improper citation of figures. There is no correlation between the figures and explanation.
· “The linear response of the stress-strain curve aspect (Figure 6b) confirms the elastic behaviour of the sample and the slope of the curve was 29 GPa for the control sample with SP (CTRL-SP), 42 GPa for the control sample with CaO (CTRL-CaO) and between 23.7 and 35.3 GPa for the samples with CNF.” Add separate bar chart for that explanation.
· Results should be supported with more reasons.
· Correlation of SEM with obtained results must be carried out.
· Conclusion must include important findings only.
Reviewer 4 Report
In this study, the Carbon nanofibers grown in CaO were investigated to determine their potential to be used as sensors in structural health monitoring (SHM).
This is a good study and its publication will be beneficial for the scientific community and will add to the existing knowledge in the fabrication of carbon nanofibers on a tailored substrate to promote self-sensing in cementitious materials.
However, some changes are required in order to be published.
- please provide the condition for the SEM and TEM determination (how the sample is prepared, what is the acceleration voltage, the spot, etc. ).
- the mix proportion is shown in Table 1, not in Table 2.
- the dimensions for the mould should be in mm not in mm3
- for the compaction please provide the conditions for the electric vibrator (time, frequency, etc).
- in table 1 the composition should be presented in per cent, not in grams.
- the electrical resistance is recorded whit a digital multiparameter presented in Figure 3B, not Figure 2B.
- in order to confirm the magnetic properties of nanoparticles present in CNF please provide VSM.
- also the chemical composition of CNT is important. The authors established "At this temperature, the reduction of Ca2Fe2O5 by ethanol results in the formation of Fe0, Fe3C, CaFe2O4 amorphous carbon, and CNF". In order to confirm this please provide an XRD analysis. CNF was purified?
- TEM is presented in Figure 4, not in Figure 3. Please correct all the comments.
- in my opinion, in figure 5, is much useful to present the DTA curves instate of the weight loss rate curves. The thermic effects can be useful to explain some reactions associated with weight loss.
- SEM is presented in Figure 6, not in Figure 5F. Please correct all the comments.
- The electrical resistivity is shown in Figure 7A. Please correct all the comments.
- "the control mortar without CNF (Figure 8A), as well as the sample with only CaO and mixed iron and calcium oxides (Figure 8B)..." I think the figure is 9, not 8.
- the strain versus FCR curves are presented in Figure 10, not in Figure 7
Reviewer 5 Report
Although the study is very interesting to the nanomaterials community and can be considered for publication, there are major issues to be addressed before consideration:
1. ICP can be performed so as to determine the residual Fe and Ca prior and post acid treatment.
2. Better quality higher resolution TEM micrographs are recommended.
3. What was the crystallinity of the Ca2Fe2O3 catalysts? Additionally, was the catalysts bimetallic or york-shell? Thus, TEM images as well as EDX of the catalysts prior to growth of CNFs must be presented, as authors claim the formation of the iron oxide composite.
4. From figure 4f, the samples are more of MWCNTs than CNFs, therefore is it still correct to say CNFs were produced in this study? This generates major confusion.
5. Techniques such as Raman analysis must performed in order to be sure whether the carbonaceous materials are MWCNTs or CNFs, especially considering the 100-500 cm-1 spectral region.
6. As the particle size of the nanocatalysts play a major role in controlling the size of the CNFs, as well as influencing the growth of the CNFs, what was the size of the Fe/CaO in this case?
7. Typically, cement contain contain Fe, Al, Mg, etc, so what guarantee do the authors that the magnetism was due to the CNFs and not from the Fe component of the cement? Also what was the elemental composition of the cement used in this study?
8. From the as-synthesized carbon-based samples, Mossbauer analysis must be done in order to determine which magnetic domains are dominant.
9. Authors claim that figure 3D shows bamboo-like fibres, however, the micrographs seem to show more twisted fibres that are covered with amorphous-mass. Could authors be careful with the use of correct terminology.
10.Usually acid treatment is known to remove metal residues as well as amorphous carbon and/or unreacted carbon species, thereby resulting in more clean samples. How come in this case, the samples still contain so much amorphous carbon after acid-treatment? Also what acid mixture was used for the treatment?
11.TGA isn't used to quantify CNF content, but rather metal content after full combustion/oxidation of carbon. Once again, correct terminology is key.
12. For the acid-washed samples, decomposition profile for the carbon core (@ 600 oC, figure 5B) shows a sudden increase in mass after complete decomposition, can authors explain this observation in detail.
13. Arrows in figure 6 are confusing as they are pointing at blank spaces. Rather include the compositions in the text. Furthermore, what does each SEM micrograph represent?
14. How do the self-sensing properties compare to other studies in literature?
Round 2
Reviewer 3 Report
Authors have addressed all the queries. Article may be accepted in the present form.
Reviewer 4 Report
The authors have improved the paper in accordance with my suggestions.The article can be accepted in the present form.